# Super-Suppression of Long-Wavelength Phonons in Constricted Nanoporous Geometries

**DOI:** 10.3390/nano14090795

**Published:** 2024-05-03

**Authors:** P. Alex Greaney, S. Aria Hosseini, Laura de Sousa Oliveira, Alathea Davies, Neophytos Neophytou

**Affiliations:** 1Department of Mechanical Engineering, University of California Riverside, Riverside, CA 92521, USA; shoss008@ucr.edu; 2Department of Chemistry, University of Wyoming, Laramie, WY 82071, USA; laura.desousaoliveira@uwyo.edu (L.d.S.O.); adavies2@uwyo.edu (A.D.); 3School of Engineering, University of Warwick, Coventry CV4 7AL, UK

**Keywords:** thermal conductivity, nanoporous materials, molecular dynamics simulations

## Abstract

In a typical semiconductor material, the majority of the heat is carried by long-wavelength, long-mean-free-path phonons. Nanostructuring strategies to reduce thermal conductivity, a promising direction in the field of thermoelectrics, place scattering centers of size and spatial separation comparable to the mean free paths of the dominant phonons to selectively scatter them. The resultant thermal conductivity is in most cases well predicted using Matthiessen’s rule. In general, however, long-wavelength phonons are not as effectively scattered as the rest of the phonon spectrum. In this work, using large-scale molecular-dynamics simulations, non-equilibrium Green’s function simulations, and Monte Carlo simulations, we show that specific nanoporous geometries that create narrow constrictions in the passage of phonons lead to anticorrelated heat currents in the phonon spectrum. This effect results in super-suppression of long-wavelength phonons due to heat trapping and reductions in the thermal conductivity to values well below those predicted by Matthiessen’s rule.

## 1. Introduction

Heat management and the control of heat flow are a cornerstone of a large number of applications and technologies. Heat-generating electronic devices require efficient heat sinks of high-thermal-conductivity materials, while energy-harvesting thermoelectric materials require ultra-low thermal conductivities to increase the efficiency of the conversion process [1]. For the latter, nanoporous materials have been very successfully used on many occasions to provide drastic reductions in thermal conductivity [2]. Even traditionally inefficient thermoelectric materials like Si can become highly efficient when they are nanostructured [3,4] or when nanopores are introduced into the lattice [5]. The size of the nanostructured features and the distance between them determines the scattering specifics, typically scattering phonons of similar mean free paths (MFPs) more effectively [6,7,8,9]. For example, short MFPs are effectively scattered by point defects or alloying; medium MFPs by nanopores, nanoprecipitates or superlattices; and long MFPs by large-scale grain boundaries in polycrystalline materials. Additionally, although Mie scattering has been shown to play a role in semiconductor materials due to point defects [10], defects of size comparable to the wavelength of phonons are generally required for (strong) Mie scattering to take place. As a result, deliberately scattering large-MFP phonons typically requires the engineering of mesoscale defects, such as grain boundaries, because of the long-wavelength nature of large-MFP phonons. However, such defects often function as recombination centers and are therefore detrimental to electron transport [11]. On the other hand, in nanostructures or nano/polycrystalline materials, the long-wavelength, long-MFP phonons—which are known to carry a large part of the heat due to their large group velocities and weaker scattering—are typically less suppressed compared to other modes [12]. In fact, many computational works by us (which will be discussed below) and others show that long-wavelength modes are less affected by material defects compared to the rest of the phonon spectrum. This is the case in typical wave-based phonon-description formalisms such as atomistic molecular dynamics (MD) or non-equilibrium Green’s functions [13]. Even the prominent grain-boundary scattering models, for example, particle-based Monte Carlo (MC) methods, or diffusive versus specular scattering models, are wavevector-dependent, with the boundary scattering strength or diffuse scattering preference favoring large wavevectors (short wavelengths) rather than the smaller wavevectors (large wavelengths) [14,15].

In Refs. [16,17], using equilibrium molecular dynamics (EMD) simulations, we presented a specially designed nanoporous structure of only a small porosity percentage, which allows strong reductions in thermal conductivity. Reductions even up to 80% larger than one would have expected from Matthiessen’s rule, taking into account the pore geometry and distances, can be achieved. We showed that this effect is a result of anticorrelations that appear in the heat current flux. Essentially, heat can be trapped between the pores in such a way that it annihilates itself and effectively leads to drastic phonon MFP reductions. In this work, we revisit this effect. We provide a summary of that observation and the special nanoporous structure design that allows for it. We then advance our understanding of this effect by focusing specifically on the behavior of long-wavelength phonons. We show that in the geometries we describe, we can super-suppress those more resilient long-wavelength, long-MFP phonons, again largely beyond what Matthiessen’s rule suggests. Such large reductions in thermal conductivity with minimal structural changes to the material can be largely beneficial to the field of thermoelectrics, for example, in which case the reduced thermal conductivity in a material that retains high electrical conductivity would lead to large improvements in important performance metrics.

The paper is structured as follows. In Section 2, we provide the background information showing the general trend of weak suppression of long-wavelength phonons compared to other modes in the presence of nanoscale defects. In Section 3, we provide details of our EMD simulation and describe the structures that show anticorrelated heat-current behavior. We further provide explanations for this phenomenon. In Section 4, we perform a wavepacket analysis as well as a power-spectrum analysis of the heat-current autocorrelation function of the heat current and show the signatures of heat confinement and phonon oscillation in the regions between the pores. In Section 5, we develop a Monte Carlo model to recreate the anticorrelation effects and compute the suppression function to investigate how the pores affect the phonon spectrum and allow for super-suppression of mean free paths and thermal conductivity. Finally, in Section 6, we conclude.

## 2. Weak Suppression of Long-Wavelength Phonons

We start by demonstrating the overall resilience of the long-wavelength phonon modes to lattice imperfections. To this end, we have simulated a series of simple nanoporous silicon geometries (Figure 1) using the fully wave-based non-equilibrium Green’s function (NEGF) formalism. The geometries as shown in the sub-figures contain a single pore or two pores. The channel has fixed boundary conditions surrounding it, i.e., we simulate a nanowire slab with open boundary conditions at the left/right sides. A complete study and full set of data are presented in Ref. [13], whereas here, we focus on the behavior of the long-wavelength phonons.

In the first example (Figure 1a), we increase the diameter of the pore from *D* = 2 nm to *D* = 6 nm and subsequently decrease the ‘line of sight’ [18]. In the other two examples, shown in Figure 1b and Figure 1c, we change the separation of the two pores in the transverse and longitudinal directions, respectively. We then compare the transmission of the different frequency modes between the pristine material and the porous materials, as computed from NEGF. As shown in all three cases in Figure 1a–c, the transmission is in general suppressed for the larger frequency modes; however, the long-wavelength, low-frequency acoustic modes are not affected as strongly (highlighted region to the left of each sub-figure). The greater resilience of these modes persists despite the large pores and the very narrow line of sight that we enforce in the simulation domain [13]. This result is in contrast to ray-tracing Monte Carlo solutions to the particle-based Boltzmann transport equation (BTE) [14], for example, where the narrow line of sight reduces the transmission of all phonon modes indiscriminately (this is shown by the solid-flat lines in the Figure 1 sub-figures). Note that here, we consider only ideal transport conditions and ignore any effects of phonon-phonon scattering or mode mixing. Umklapp scattering typically affects high-energy phonons more; thus, the transmission of high-energy phonons is suppressed compared to that of acoustic long-wavelength phonons, to begin with. However, the main point is that the long-wavelength modes are highly resilient to nanostructuring compared to higher-frequency modes.

## 3. Geometries, Methods, and the Anticorrelation Effect

*Geometries and methods:* The structures we model in equilibrium molecular dynamics (EMD), from here onwards, have simplified geometries consisting of cylindrical pores placed in arrays perpendicular to transport, as shown schematically in Figure 2, and extending to the entire depth of the material.

We consider a system of up to 108 nm in length, large enough to allow over 85% of long-wavelength modes to transverse the domain [19]. The thermal conductivity within EMD is computed via the Green–Kubo method as follows:(1)$$\kappa_{x}=\frac{V}{k_{B}T^{2}}\int_{0}^{\infty }\langle J_{x}\left(t\right)J_{x}\left(t+\tau \right)\rangle d\tau$$
where V is the volume of the periodic simulation cell (the volume of Si plus the volume of the pores), T is the temperature, $k_{B}$ is Boltzmann’s constant, $J_{x}\left(t\right)$ is the heat flux in *x*-, and $\langle J_{x}\left(t\right)J_{x}\left(t+\tau \right)\rangle =A(\tau )$ is the averaged heat-current autocorrelation function (HCACF) at interval τ.

For our EMD simulations, we used the LAMMPS software (29 October 2020 stable version) [20], and the Stillinger–Weber (SW) potential [21], which is commonly used for heat transport in silicon by us and others [12,16,17,22,23], as it provides a good estimation of the phonon dispersions, despite overestimating the thermal conductivity to values of ~250 W/mK [24]. Simulation domain dimensions of up to ~108 nm in length and ~5.4 nm in the width were employed, while the thickness of the domain was ~5 nm in all cases. We typically simulate the same system up to 20 times in order to average the results and reduce the uncertainty in Green–Kubo [18]. The simulations were performed at ~300 K, and the transport properties are computed along the *x*-axis in the [100] crystal orientation.

*Anticorrelation effect:* We now introduce the heat−current anticorrelation effect that is central to this paper and the different geometrical features that control it. We consider four porous geometries, as depicted in Figure 3a. The moving average of their HCACFs and the HCACF cumulative integrals from EMD are shown in Figure 3b and Figure 3c, respectively, with the same colors as the corresponding geometries in Figure 3a.

Typically, the HCACF for Si decays exponentially and monotonically to zero [25] (ignoring the expected statistical noise [18,26]), with the decay rate being a measure of the relaxation times of the propagating phonons. The first geometry of Figure 3a (cyan) with pore radius 1 nm, as we show in Figure 3b, matches this behavior. It also shows the normal increase and saturation to the cumulative HCACF for the thermal conductivity in Figure 3c (cyan line).

However, the HCACFs of the rest of the three porous geometries with narrow necks show *anomalous* behavior. They show an *anticorrelation* trend, where the HCACFs turn negative and then increase again to approach the *x*-axis from below. This behavior causes a peak in the cumulative HCACF, as shown in Figure 3c, after which a downward trend is observed, reducing the thermal conductivity of the material (see the second geometry in gold color as a basis; in that case, we increased the size of the pores slightly compared to the first structure). 

The decline in the HCACF value and the drop in its cumulative integral can be further increased by increasing the pore diameter and reducing the neck size (green structure and data lines). In this case (green structure), a larger decrease in the HCACF values is achieved, and this change enhances the reduction in the thermal conductivity in Figure 3c. In fact, we find that the AC effect is better controlled by the neck-to-pore-radius ratio than by either of the two quantities alone [16,17], as the combination of both quantities reflects phonons more effectively. We also find that beyond a certain point, corresponding to ~80% thermal conductivity reductions, reducing the neck does not enhance the AC effect. Possibly, a limit is reached at which point the remaining phonons have mean free paths smaller than the pore periodicity—see Figure 2), which will thermalize before anticorrelations form. This effect suggests that this drastic reduction in thermal conductivity is a result of super-suppression of the long-mean-free-path long-wavelength phonons. The fact that the ratio of the neck radius to the pore radius is what determines the AC is important because it can allow for this effect to be observed for larger neck/pore sizes, which could be more easily realized [17]. We can quantify the reduction in κ due to AC by considering the distance of the peak of the cumulative HCACF integral from its final converged value. For example, for the geometry shown in green in Figure 3c, this value corresponds to a decrease in the accumulated κ of approximately 52%. This value is an underestimation of the real reduction in thermal conductivity, as in a geometry of similar porosity, but without the AC effects, the cumulative HCAF will continue to increase beyond the point of the peak in the AC-producing geometry before it eventually saturates.

The anticorrelation effect can also be controlled on the *x*-axis, i.e., by controlling the correlation length of the phonons that contribute to the effect by increasing the period of the pore array (orange structure compared to the gold) in the transport direction. This change shifts the HCACF minimum further to the right at higher correlation times (Figure 3b), while the peak of the HCACF cumulative integral (Figure 3c) moves to the right. The overall thermal conductivity is increased compared to the same structure with shorter periodicity, as expected.

We find that there is a linear correlation between the pore periodicity, *d*, and the points when the anticorrelation minima occur, and this correlation is related to the sound velocity of phonons travelling between the pore regions. The fact that the location of the drop in values is related to the sound velocities of the acoustic phonons is also an indication that the anticorrelation effect primarily targets those modes.

Finally, we note that many other works in the literature have reported oscillatory or anticorrelation behavior in the HCACF from Green-Kubo simulations, particularly in systems with more than one element. A few examples are metal-organic frameworks (MOFs) [27,28]; organic crystals [29,30,31]; liquids in which convective atomic motion (i.e., mass transport) results in negative HCACF minima [32]; layered materials with large atomic mass differences, as in HfB_2_ [33]; and amorphous materials [25,34], in which case McGaughey and Kaviany attribute this behavior to distinct local environments. In all these examples, the mechanisms responsible for anticorrelations are distinct from the diffusive backscattering of the heat flux in the crystalline close-packed nanoporous geometries presented in this work. For example, in our simulations, the material is crystalline, consists of only a single element, and has a fixed center of mass, so convective heat flux is negligible and only the virial term contributes to the thermal conductivity. Moreover, in these examples, the HCACF oscillations have sub-picosecond periods, whereas the drop in HCACF values in the crystalline porous geometries we study here is long-lived (tens of picoseconds). The report of anticorrelated heat-current fluctuations that Haskins et al. [35] in carbon nanotubes (CNTs) is the only study that we have found related to ours, as the authors attributed this behavior to the elastic scattering reflection of the heat flux at the boundary.

## 4. Heat Trapping: Wavepacket and Power Spectrum Analysis

*Wavepacket analysis and heat trapping:* We have shown that the AC effect can strongly reduce thermal conductivity with limited changes to the material compared to the typical effects that involve many scattering centers and large defect surface area. This effect arises from the efficient entrapment of heat between the pores, where phonons undergo reflection before thermalization, thus strongly reducing their overall mean free path. To further illustrate the heat trapping that leads to the AC effect and super-suppression of the MFPs and thermal conductivity, we have modeled the propagation of phonon wavepackets in porous Si and monitored their trajectory.

Wave packets were constructed by preparing the system with the initial atom displacements and velocities arising from a linear superposition of plane waves weighted by a Gaussian distribution around a localized wavevector. In this scheme, the displacement ($u_{lj\mu \gamma }$) of the *j*th atom in the *l*th unit cell along direction μ (*x*, *y* or *z*) due to a wave packet of phonon mode γ is given by the following equation [17]:(2)$$u_{lj\mu \gamma }=\sum_{q}^{\;}A_{o}\left(\frac{1}{\sigma \sqrt{2\pi }}\right)e^{{\left(\frac{\left|q-q_{\gamma }\right|}{\sigma \sqrt{2}}\right)}^{2}}\epsilon_{j\mu q}e^{-i\left(r_{l}\cdot q+\omega_{q}t\right)},$$
where $r_{l}$ is the position vector of the *l*th unit cell, $q_{\gamma }$ is the wave vector of the carrier wave and $A_{o}$ and σ are the wavepacket’s amplitude and momentum uncertainty, respectively. The summation is performed over all modes that are in the same acoustic branch as mode γ and that have wave vectors *q* that are commensurate with the simulation cell. The terms $\epsilon_{j\mu q}$ and $\omega_{q}$ are the eigenvector and angular frequency of mode with wave vector *q*, respectively. During preparation of the simulation, the initial displacements were computed using t=0 and the initial velocities were computed from the time derivative of Equation (2).

We set a wavepacket to propagate through a porous Si slab with pore radii of 2 nm and corresponding 1.4 nm neck, a system that shows AC effects. Heatmaps show the evolution of the kinetic energy of the wavepackets along the material for different values, as indicated in each sub-figure (Figure 4). The x-axis shows the propagating direction in this geometry, with the pores located at the vertical white lines 54 nm apart. The y-axis shows the time evolution of the wavepacket kinetic energy.

Figure 4 shows that heat, upon encountering the pores, is partially reflected and partially transmitted. Due to the narrow spacing between the pores, phonons (wavepackets here) get trapped and oscillate in the region between the pores. Heat is trapped for all wavevectors. However, note that the wavepackets are only weakly affected by Umklapp scattering, as we consider a very low temperature, 5 K, in these simulations. At room temperature, due to their stronger Umklapp scattering, large wavevector/frequency phonons will thermalize before they propagate fully in the region between the pore stacks. Thus, this oscillation is more relevant to long-mean-free-path phonons at elevated temperatures, which will bounce a few times instead of multiple times, as we show below. This trapping of low wavevectors that is observed here for the larger pores/smaller necks leads to drastic reductions in thermal conductivity.

*HCACF power spectrum analysis:* To further illustrate indications of heat trapping and super-suppression of long-mean-free-path phonons, we now perform an analysis of the spectrum of the HCACF. Various nanoporous geometries (see Figure 5a) were simulated with EMD, as described above. These geometries included pores—which are drilled through the entirety of the slab (left column)—and voids, which are small cavities within the slab (right column). The geometries of these systems varied from uniform placement to staggered, clustered, and randomized placements. Pore size and distance have also been varied between systems (see Appendix A). The colors of the lines plotted in Figure 5b,c correspond to the similarly colored geometries in Figure 5a.

In Figure 5b, the thermal conductivity calculated via the Green-Kubo formalism has been plotted for each of the nanoporous systems, as well as for pristine silicon, in black (only for very small correlation times, as it saturates at much higher values compared to the other geometries). A plot showing the full range of the pristine thermal conductivity and the HCACF is shown in the Appendix A. The first group of points around a porosity of 2% are the first three pore geometries shown in the left column of Figure 5a. Despite the low porosity, a significant reduction in thermal conductivity from the pristine to the porous systems is highly evident (the black line saturates around 250 W/mK, see Appendix A—note that it is well known that the Stillinger−Weber potential overestimates the thermal conductivity of Si). What is interesting, however, is that the system in this porosity group with the largest but fewest pores (i.e., with fewer scatterers per volume) reduces the thermal conductivity the most (bold dark orange line). This effect can be largely attributed to presence of the AC effect arising from that particular backscattering of phonons, as described above. The next geometry group we consider is in the right column of Figure 5a, with around 5% porosity; this geometry consists of the four void geometries. The same behavior is witnessed for these in Figure 5b, where thermal conductivity is reduced. The same anticorrelation effect is present, but to a much lesser degree compared to the pore geometries. As voids are not drilled through the entire slab, there are pockets of silicon space around each void where phonons can continue propagating, resulting in fewer backscattering events and less phonon trapping.

Now we go back to the first column for the third porosity group of around 10%, which consists of the last three pore geometries in Figure 5a. Here we see an even more significant reduction in thermal conductivity in Figure 5b, depending on the size of pores and the distance separating them. In the dark blue geometry, we see the most pronounced anticorrelation effect and the greatest reduction in thermal conductivity. This occurs despite the presence of fewer pores and thus greater distances between the pores, and expectably a larger characteristic scattering length, defined by the distance between the pores along the transport (*x*-axis) direction. The insert in Figure 5c shows these thermal conductivity values plotted against porosity for each system. Notice that drastic reductions in thermal conductivity are achieved with very small porosities—typically much larger porosities are needed in common geometries with no AC effects [14]. Therefore, pore size and distribution are not the only effects playing a role in thermal conductivity reduction, as evidenced by the different results at equal porosities, but also the distance between pores, both along and perpendicular to the direction of heat transport.

Figure 5c shows the HCACF power spectrum versus the power-spectrum frequency—here, we focus on the left side of the spectrum, on the low frequencies, for reasons to be explained further below. The power spectrum of the HCACF of phonons that follow the expected Poisson distribution of lifetimes typically drops with frequency by *f^−^*^2^, signaling Brownian motion and resulting in finite heat flux (see the log-log plot in Figure 6a below for the bulk-black line and the AC devices in dark blue and dark orange). In general, in the presence of pores, in most cases, the entire power spectrum reduces in amplitude and its amplitude becomes more uniform across different power-spectrum frequencies, while a downward trend is typically retained as the frequency increases.

In the case of structures with AC effects (dark blue and dark orange cases), however, additional peaks arise in the power spectrum; these are more pronounced for the dark blue line, with the strongest AC in its HCACF. These peaks would correspond to a resonance (e.g., Gaussian or Lorentzian distributions of lifetimes) built on top of the Poisson distribution. In contrast to Poisson distributions, such resonances lead to zero net heat flux. It would be another indication of phonon-reverberation processes between the pores. Essentially, trapped phonons will undergo backward and forward reflections with no contribution to the heat flux. The rise in intensity in those regions of the power spectrum indicates that a large portion of the heat is trapped in these reverberate oscillations. The majority of those would be long-wavelength phonon modes with mean free paths that are long compared to the separation between the pore stacks—long enough to undergo (multiple) reflections and finally thermalize somewhere in between the pore stacks, suppressing their effective mean free path. The additional peak for these geometries is also evident in the log-log plot of the power spectrum in Figure 6a, where it deviates from the *f*^−2^ behavior, indicating a preferable frequency where the system oscillates.

In Figure 6b, we provide a rough first-order estimate of how the power-spectrum frequencies of Figure 5c can be related to corresponding length scales in the system. To this end, we proceeded as follows: we calculated the average speed of the acoustic phonons using the Stillinger−Weber (SW) potential to be *υ*_ac_∼6033 m/s [16]. By multiplying the *τ*^−1^ variable in the *x*-axis of the spectra in Figure 5c by *υ*_ac_^−1^, we can obtain an estimate of an inverse length *l*^−1^, which is related to the path that acoustic phonons will travel before thermalizing. Figure 6b is a repetition of Figure 5c, but with the x-axis changed to this length. The different-colored lines correspond to the same lines in Figure 5c and the geometries of Figure 5a. For the specific region of the peak in the power spectrum, this will constitute the characteristic reverberation-length scale (as we name the *x*-axis). We show the spectra for the long lengths, i.e., those longer than *l*~80 nm (from the right side of the *x*-axis towards the origin at the left side). The geometry that shows the super-suppression of thermal conductivity due to the AC effect (thick blue) has its peak centered around ~225 nm. This result indicates a characteristic length-scale behavior of phonons in the system. The distance between the pores in the propagating direction is 54 nm; thus, this value could correspond to four periods of phonon travel (e.g., two back-and-forth reflections between consecutive pore stacks or one back/forth reflection in two pore-stack regions). At this point we don’t offer any further analysis on this, the side peaks (whether they correspond to more/less number of oscillations), or even separate the effect of pores on longitudinal or transverse phonons with different sound velocities, since this this provides only a rough estimate.

## 5. Phonon-Suppression Function within a Monte Carlo Model

To elucidate how AC arises from heat trapping between pores, we have developed a statistical model for a gray population of phonons at equilibrium. The model is described in detail in Ref. [17] but is briefly outlined again here. The instantaneous heat flux in a volume V of crystal at thermal equilibrium with no net temperature gradient arises from the stochastic occupancy of the phonon modes in the volume, so that
(3)$$J_{x}\left(t\right)=\frac{1}{V}\sum_{i}^{\;}n_{i}\left(t\right)\;j_{i},$$
where $j_{i}=\omega_{i}\hbar \;\left(v_{i}\cdot \hat{x}\right)$, is the energy current along $\hat{x}$ due to a single phonon in the *i*-th phonon mode. Here ℏ is the reduced Plank constant and $\omega_{i}$ and $v_{i}$ are the phonon mode’s angular frequency and group velocity, respectively. The instantaneous occupancy, $n_{i}\left(t\right)$, in Equation (3) is the number of phonons exciting the *i*-th mode at time t, and the summation is performed over all vibrational degrees of freedom (phonon modes) in the volume. 

The key insight of our model is that while the instantaneous heat flux depends on the superposition of the occupancy $n_{i}\left(t\right)$ in all modes, one can decompose the total autocorrelation function into only the sum of correlation functions of individual phonon modes’ deviations from their mean Bose-Einstein occupancy $\Delta n_{i}\left(t\right)=n_{i}\left(t\right)-\langle n_{i}\rangle$ if one assumes there to be no correlation between the initial and scattered phonon mode(s) in each intrinsic scattering event. This approximation is equivalent to the relaxation time approximation (RTA) and results in an expression for the HCACF as follows:(4)$$\langle J_{x}\left(t\right)J_{x}\left(t+\tau \right)\rangle =\frac{1}{V^{2}}\sum_{i}^{\;}\frac{j_{i}^{2}}{\langle \theta_{i}\rangle }\langle \Delta n_{i}^{2}\rangle {\langle A\left(\tau ,\theta \right)\rangle }_{\;},$$
where $\langle \theta_{i}\rangle$ is the i-th mode’s average phonon lifetime, $\langle {\Delta n}^{2}\rangle$ is the variance of the occupancy, and $A\left(\tau ,\theta \right)$ is the autocorrelation function of the boxcar function $\Pi_{0,\theta }\left(t\right)$ with width θ, so that
(5)$$A\left(\tau ,\theta \right)=\int_{0}^{\infty }\Pi_{0,\theta }\left(t’\right)\Pi_{0,\theta }\left(t’+\tau \right)dt=\left\{\begin{array}{c} \theta -\tau \;for\;0\leq \;\tau \leq \theta \;\\ 0\;otherwise \end{array}\right..$$

The variance of the occupancy is computed by the summation over fluctuations weighted by the probability $P_{n}=e^{-n\frac{\omega \hbar }{k_{B}T}}\left(1-e^{-\frac{\omega \hbar }{k_{B}T}}\right)$ of finding a mode in state of occupancy n and is found to be proportional to the derivative of the average occupancy with respect to temperature, as follows:(6)$${\langle {\Delta n}^{2}\rangle }_{\;}=\sum_{n=0}^{\infty \;}P_{n}{\Delta n}^{2}=\frac{e^{\frac{\omega \hbar }{k_{B}T}}}{{\left(\frac{\omega \hbar }{k_{B}T}-1\right)}^{2}}=\frac{k_{B}T^{2}}{\omega \hbar }\frac{d\langle n\rangle }{dT}.$$

The average of the excitation autocorrelation functions, $\langle A\left(\tau ,\theta \right)\rangle$, is calculated over the Poisson distribution of $P_{\theta }=\frac{1}{\langle \theta \rangle }e^{-\theta /\langle \theta \rangle }$ of random waiting times for a stochastic process, as follows:(7)$${\langle A\left(\tau ,\theta \right)\rangle }_{\;}=\int_{0\;}^{\infty \;}P_{\theta }\;A\left(\tau ,\theta \right)\;d\theta =\langle \theta \rangle \;e^{-\tau /\langle \theta \rangle }.$$

Substituting Equations (6) and (7) into (4) leads to an expression for the HCACF as follows: (8)$$\langle J_{x}\left(t\right)J_{x}\left(t+\tau \right)\rangle =\frac{k_{B}T^{2}}{V}\;\sum_{i}^{\;}C_{i}{\left(v_{i}\cdot \hat{x}\right)}^{2}\;e^{-\tau /\langle \theta_{i}\rangle },$$
where $C_{i}$ is the i-th vibrational mode’s contribution to the volumetric specific heat. When Equation (8) is used in the Green-Kubo formula in Equation (1), it yields the expression for the thermal conductivity derived from the Boltzmann transport equation using the RTA.

The analysis above relies on decomposing the heat current into independent fluctuation events that are self-correlated but not cross-correlated with one another. The addition of pores to a crystal introduces extrinsic backscattering of phonons at pore surfaces. In this process, when a phonon is backscattered from mode $i\to i^{’}$, the heat current $j_{i^{’}}$ from the backscattered phonon is correlated with the heat current $j_{i}$ from the incident mode because, even in the case of diffuse scattering where the direction of the scattered phonon is independent of the incident phonon, at least the sign of the heat flux normal to the scattering surface is reversed by backscattering, making $j_{i}$ and $j_{i^{’}}$ anticorrelated. In this scenario, the independent self-correlated events that make up the total HCACF are still the period between intrinsic (uncorrelated) scattering events, but the heat current during an event now consists of a sequence of discrete values as the phonon is backscattered between different phonon modes. The HCACF from an individual trajectory with a sequence of multiple backscattering events, $A_{b}(\tau )$, is a linear piecewise function that is straightforward to compute numerically.

We have developed a simple ray-tracing scheme to simulate the trajectories of phonons undergoing defuse elastic backscattering and compute their HCACF. The code performs Monte Carlo sampling over trajectories and lifetimes to compute the average HCACF $\langle A_{b}(\tau )\rangle$ from an isotropic gray phonon gas. The details of this model were described in a prior work [17]; but here, we apply the model to compute the phonon-suppression function to determine how pores affect the phonon spectrum’s contribution to the total thermal conductivity. We used the model to compute the HCACF for a phonon gas with mean free path λ over a broad sweep of the Knudsen number from the diffusive to ballistic regimes. The Knudsen number is defined as the average of the MFPs of gray phonons over the geometrical distance between the pores, $K_{n}=\lambda /d$. This calculation was done for both the staggered and aligned pore geometries in the inset of Figure 7a, with a Gaussian process regression used to construct an interpolation function $\tilde{A}\left(K_{n},\tau \right)=\langle A_{b}\left(\tau \right)\rangle$ for the mean normalized correlation function of the backscattering phonon gas as a function of Knudsen number.

To compare the MC model against the MD simulations, we used AlmaBTE [36] to compute the phonon lifetimes for Si using the 2nd- and 3rd-order interatomic force constants for Si modeled with the Stillinger−Weber empirical potential. The distribution of phonon properties computed on a 30×30×30 q-point Brillouin zone mesh where spherically symmetrized and used to compute the HCACF for a “multi-gray” model of monolithic and nanoporous SW Si using the following equation:(9)$$\langle J_{x}\left(t\right)J_{x}\left(t+\tau \right)\rangle =\frac{k_{B}T^{2}}{V}\frac{1}{3}\sum_{i}^{\;}C_{i}\;{\left|v_{i}\right|}^{2}\;\tilde{A}\left(\frac{\lambda_{i}}{d},\tau \right),$$
the results of which are shown in Figure 7a. For fair comparison to the MD simulations where mode occupancy is classical, the classical values of $C_{i}$ were used in Equation (9). This model has no free-tuning parameters, and although the intrinsic phonon lifetimes predicted by AlmaBTE are longer than those observed in MD (which include four and more phonon interactions), its overall match of the multi-gray MC model to the MD HCACF is qualitatively very good. The MC-computed HCACF shows clear anticorrelation for aligned pores that is absent for staggered pores, as observed in the MD as well. 

Having established the validity of the ray-tracing model, we used it to compute the thermal conductivity *suppression function* $S\left(K_{n}\right)$ for the phonon gas in these two pore layouts with the following equation:(10)$$S\left(K_{n}\right)=\frac{\phi }{\langle \theta \rangle }\int_{0}^{\infty }\tilde{A}\left(K_{n_{i}},\tau \right)\;d\tau ,$$
where ϕ is the volume fraction of the pores. The suppression function indicates how much the pore geometry suppresses the thermal conductivity. The suppression functions for the staggered and aligned pores are shown in Figure 7b and Figure 7c, respectively. The suppression in the diffusive regime (low $K_{n}$) originates entirely from the reduced number of vibrational degrees of freedom as a result of the porosity and is the same for both geometries, as they have the same ϕ. However, during the crossover to the ballistic regime, the aligned pores exhibit a steeper suppression of the thermal conductivity (lower blue line in the inset of Figure 7b). To better understand this transition, we fitted the suppression function with a Matthiessen rule (MR) model as follows:(11)$$S_{MR}\left(K_{n}\right)=S_{V}{\left(\frac{1}{\lambda }+\frac{1}{\beta_{d}}\right)}^{-1}=S_{V}\frac{\beta }{K_{n}+\beta },$$
where $S_{V}=(1-\phi )$ is the suppression coming from the missing material at the pores (with ϕ being the volume fraction of porosity) and $\beta_{d}$ is the effective extrinsic phonon mean-free-path for scattering at pores, with β being a dimensionless geometric factor. For the case of the staggered pores, the suppression is fit well by Equation (11) with geometric scattering factor β=0.62 (the black line in Figure 7b). This factor makes sense, as the pores are staggered and so the typical distance that a phonon must travel to impinge on a pore is on the $∼\frac{1}{2}d$. In contrast, the suppression function from the aligned pores is not fit well by Equation (11) using a single value for β (Figure 7c). The left-hand portion of the suppression plot (the diffusive regime) can be matched by Equation (11) using a value of β close to 1 (dashed black line), which makes sense, as the pores are aligned. However, to match the right-hand (ballistic) portion of the suppression plot to MR requires using a much smaller geometric factor, indicating that in the ballistic regime, the effective mean free path for scattering phonons at pores is significantly smaller than the distance between the pores. To account for this crossover in behavior, instead of using a fixed value of β in Equation (11) as would be expected by Matthiessen’s rule, we have used a scale-dependent geometric factor, as follows:(12)$$\beta \left(K_{n}\right)=\frac{K_{X}\;\beta_{d}+K_{n}\;\beta_{b}}{K_{X}+K_{n}},$$
where $\beta_{d}$, and $\beta_{b}$ are the dimensionless characteristic scattering lengths in the diffuse and ballistic limits and $K_{X}$ is the Knudsen number at the crossover between these regimes. A good fit to the suppression function from aligned pores was obtained with fitting parameters $\beta_{d}=0.92$, $\beta_{b}=0.17,$ and $K_{X}=0.19,$ as is shown by the solid black line in Figure 7c. This deviation from Matthiessen’s rule and the dramatic reduction in the geometric scattering factor from $\beta_{d}$ to $\beta_{b}$ from the diffusive to the ballistic regimes represents a five-fold super-suppression of the phonon mean free path and a more than three-fold reduction compared to the staggered case!

In the staggered geometry, where the pores are better dispersed, the average line of sight distance between pores along the transport direction is shorter, and thus, in the diffuse regime, staggered pores provide more effective thermal resistance. However, in the ballistic regime, we observe strong anticorrelation in the HCACF of the aligned pores that is not present for the staggered pores. In these MC simulations, the backscattering behavior when a phonon collides with the pores is identical in both the aligned and staggered cases; thus, the anticorrelation must arise not from individual pores, but from the collective arrangement of pores. We propose that is due to a shadowing effect from the aligned pores.

The anticorrelation in the heat current along x will be strongest from phonons that collide and are backscattered by surfaces perpendicular to x, as the anticorrelation arises only in the component of velocity normal to the scattering surface. In both systems, the MC and MD Green-Kubo simulations are performed at equilibrium, so although we call what follows a shadowing effect, the phonon radiance hitting the pore surface is the same at all points around the pore. However, the fetch that the phonons travel across before they impinge on the pore is not the same in all directions. In the aligned pores, shadowing by neighboring pores means that phonons striking the pores in the ligature sections between pores can have traveled only a short distance before hitting the pore boundary. The contribution that an individual fluctuation event (phonon) makes to the HCACF is proportional to the square of its distance of travel. Thus, the shadowing does not change the intensity of phonons striking the pores laterally but diminishes their contribution to the HCACF. This relationship means that the HCACF is composed mainly of long-fetch phonons, which also strike the pores in the direction that gives maximum anticorrelation from backscattering.

To understand which phonons are suppressed by the pores, we plot in Figure 7d the cumulative conductivity over the spectrum of intrinsic phonon mean free paths (MFPs) in Si, as well as that of the two geometries examined. While contributions from phonons with all MFPs to conductivity is reduced, the cumulative distribution becomes particularly flat for the long MFP modes in the porous geometries and more prominent in the AC geometry (blue line). To understand the correlation between phonon wavelength and backscattering, we have plotted in Figure 7e,f a scatterplot of the wave number of phonon modes in Si vs. their suppression functions from the staggered and aligned pores. The suppression function is the factor by which a mode’s contribution to conductivity is suppressed. We see that the lowest suppression factors are reached in the case of aligned pores in the low and mid q regions, which are highlighted with a blue oval. The same region in the plot for the staggered pores is empty.

Finally, we note that porosity in semiconducting materials has been widely studied both experimentally and computationally for well over two decades [9,37,38]. The introduction of nanopores has been shown to, in some cases, reduce the room-temperature thermal conductivity of semiconducting materials beyond the materials’ amorphous limit through strong phonon-boundary scattering at the pore surfaces [2,5]. Other works have shown that phonon transport and path guidance can be engineered through “ray-phononics” guided through the use of pores and pore scattering, as defined by Anufriev and Nomura et al. [39,40]. Porosity, pore spatial distribution/arrangement (e.g., distances between pores, misalignment, clustering, or randomization) [8,12,18,41,42], size [8,41,43,44], shape [43,45], number [43,44], and boundary surface area and roughness [8,12,14,43] are several of the geometrical degrees of freedom found to play a role in reducing the thermal conductivity. Most of this behavior can be ascribed to either boundary scattering or reduction of the phonon line of sight, whereas the thermal conductivity is in most cases not reduced beyond the characteristic defect length scale imposed by the distances between the pores. Thus, Matthiessen’s rule can suitably explain the reduction in thermal conductivity observed for such porous geometries, unlike that we describe in this work.

## 6. Conclusions

In this work, we have performed large-scale equilibrium molecular-dynamics simulations in special types of Si porous systems porous systems and illustrated the possibility of trapping long-wavelength, long-mean-free-path phonons between pore regions. This phenomenon results in super-suppression of their mean free paths and by extension a super-suppression of the thermal conductivity to a value far below the value Matthiessen’s rule will predict. In the simulations, this effect appears as anticorrelations in the heat-flux autocorrelation function, indicating that the scattered phonons undo their heat flux, an effect that translates into a drastic reduction of their mean free paths. We further explain this effect with wavepacket simulations, resonances in the power spectrum of the HCACF, and an equilibrium Monte Carlo model of a gray-phonon gas. All results support the claim that long-wavelength phonons and their contribution to thermal conductivity are strongly suppressed. The findings of this work would benefit applications that target strong reductions in thermal conductivity with minimal structural changes, as in the case of thermoelectric materials.

## Figures and Tables

**Figure 1 nanomaterials-14-00795-f001:**
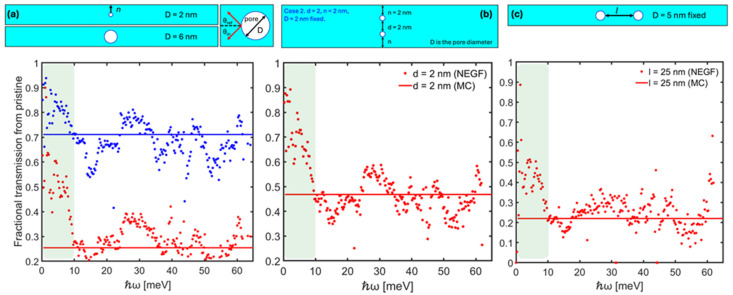
Normalized Si-nanochannel phonon-transmission functions for different phonon energies ℏω, extracted from NEGF, indicating the resilience of the long-wavelength acoustic phonon transmission to nanostructured defects (left highlighted region). The top row shows schematics of the geometries studied. The bottom row shows the transmission functions normalized to the transmission of the pristine channel. The channel has a length of 100 nm and a width of 10 nm. The flat, solid lines indicate the normalized transmission of the ray-tracing Monte Carlo solution to the BTE. (**a**) A single pore is placed in the domain with two diameter cases: *D* = 2 nm (blue) and *D* = 6 nm (red). In the latter case, the neck size, *n*, is reduced. (**b**,**c**) Pores are placed as shown in the schematics.

**Figure 2 nanomaterials-14-00795-f002:**
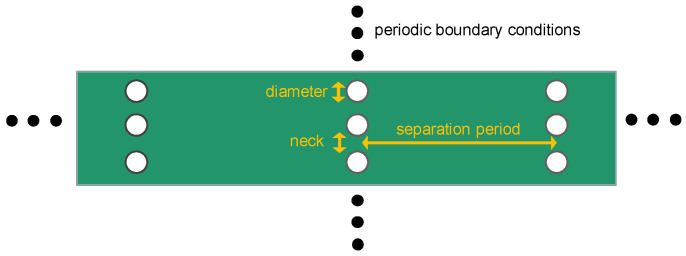
Schematic of the structure designed to allow for heat−current anticorrelations and ultra-low thermal conductivities. Pores are designed with specific separation periods in the transport direction, diameters and neck sizes. The assumed underlying material is Si.

**Figure 3 nanomaterials-14-00795-f003:**
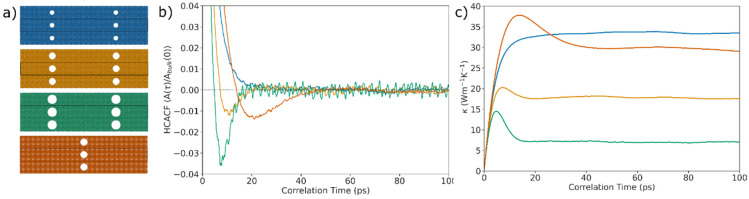
The anticorrelation (AC) effect. (**a**) The four porous Si geometries considered: one that does not show AC (blue) and three with AC. (**b**) The HCAF for the four cases. (**c**) The cumulative HCAF. The data colors in (**b**,**c**) correspond to the structure colors in (**a**). The pore radii (distances) in (**a**) are 1 nm (27 nm), 1.5 nm (27 nm), 2 nm (27 nm) and 1.5 nm (54 nm).

**Figure 4 nanomaterials-14-00795-f004:**
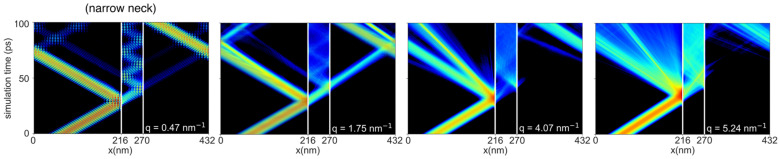
Heatmap of the time evolution of the wavepacket kinetic energies in the nanoporous geometry which shows AC effects. The pore radius is 2 nm, and the neck size 1.4 nm. The white lines indicate the pore location in the transport direction. Each plot corresponds to a wavepacket of a transverse acoustic mode centered at the wavevector, q, as indicated. The distance between the pores is 54 nm. The amplitudes of wavepackets are tuned to keep the temperature as low as 5 K in all of the simulations. Adapted with permission from Ref. [16]. Copyrighted by the American Physical Society.

**Figure 5 nanomaterials-14-00795-f005:**
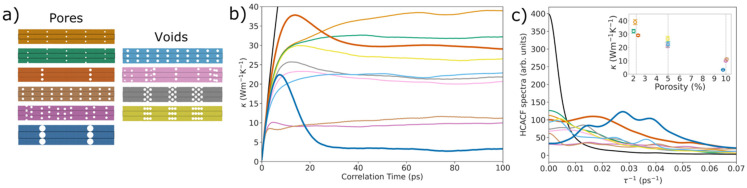
Analysis of the spectrum of the HCACF. (**a**) The geometries simulated with EMD. (**b**) The HCACF integral, or cumulative thermal conductivity. The black line indicates the corresponding bulk Si HCACF integral. In bold, we show the geometries that have strong AC effects. (**c**) The low-frequency end of the power spectrum of the HCACF. The power spectrum is plotted with a Gaussian smoothing of 100 to remove much of the accompanying noise. The data colors in (**b**,**c**) correspond to the structure colors in (**a**). Inset of (**c**): the thermal conductivity of the systems versus porosity. The blue geometry with an AC effect (and, at a lesser degree, the dark orange one), has a large peak with side peaks in the middle region of the depicted power spectrum originating from the reverberation of phonons between the pore regions. The pore radii (distances) for the geometries in (**a**) are as follows: 1 nm (20 nm), 1 nm (20 nm), 1.5 nm (51 nm), 1.5 nm (8 nm), 1.5 nm (N/A), 3.5 nm (46 nm), 1.5 nm (8 nm), 1.5 nm (N/A), 1.5 nm (1.6 nm), and 1.5 nm (1.6 nm). The distances listed as N/A are for the porous systems where distances between pores/voids are randomized.

**Figure 6 nanomaterials-14-00795-f006:**
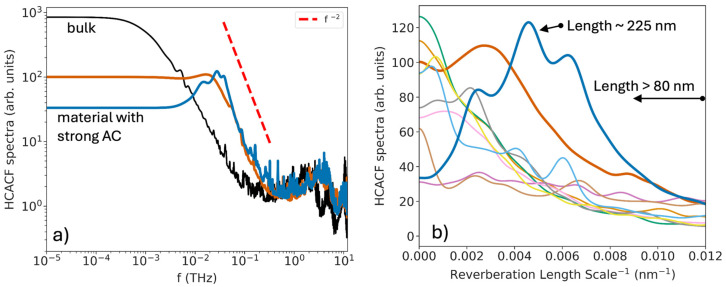
(**a**) The spectrum of the HCACF on a log-log scale for bulk Si (black line) and the AC geometries of Figure 5a, following the same color scheme. (**b**) The spectrum of the HCACF versus a first-order estimate of a length scale in the system, indicating the reverberation of phonons. The *x*-axis shows the inverse of the length scale, with the far-right region corresponding to ~80 nm. The system with the largest AC effect (thick blue line) peaks around a *length* ~225 nm.

**Figure 7 nanomaterials-14-00795-f007:**
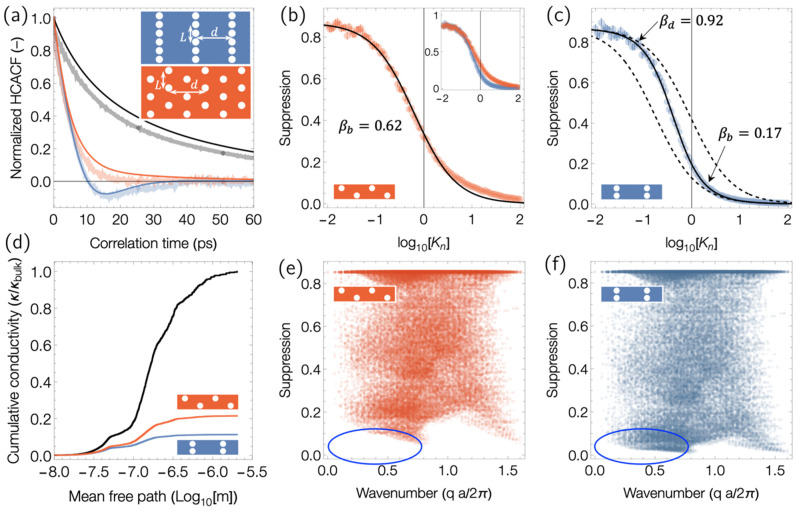
(**a**) Comparison of the normalized HCACF predicted by the multiple gray-phonon gas models of SW Si (solid lines) with those from MD (translucent lines). The gray lines are for bulk Si, while the colored lines are for the pore geometries shown in the inset, where d=54 nm, L=26 nm, and the pore radius is 5.4 nm. (**b**) Suppression function for the gray-phonon gas computed from the MC model with staggered pores and the Matthiessen rule fit with β=0.62. (**c**) The suppression function for the aligned pores showing the Matthiessen fit with a crossover in scattering length from $\beta_{d}=0.92$ in the diffuse regime to $\beta_{b}=0$.17 in the ballistic regime. The inset plot in (**b**) shows the overlay of the suppression function for staggered and aligned pores. (**d**) Cumulative conductivity distribution over a phonon mean free path normalized by $\kappa_{bulk}$. (**e**,**f**) are scatterplots of phonon wave number vs. suppression functions for staggered and aligned pores, respectively. The blue ovals indicate the long-wavelength region where strong suppression is observed from the aligned pores, whereas the corresponding region in the staggered-pore case is empty.

## Data Availability

Data are contained within the article and Appendix A.

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
