# Peer review of "Super-Suppression of Long-Wavelength Phonons in Constricted Nanoporous Geometries"

_nanomaterials, 2024, doi:10.3390/nano14090795_

Round 1
Reviewer 1 Report
Comments and Suggestions for Authors
Reviewer 2 Report
Comments and Suggestions for Authors
This paper reports a lead to anticorrelated heat currents in the phonon spectrum. This results in super-suppression of long-wavelength phonons due ot heat trapping, and reductions in the thermal conductivity using MD simulations. The phenomenon is interesting and the method is in general reasonable. The paper can be published after minor revision. I have a few comments and questions for the authors to address.
1. The actual pore size and pore distance in Fig. 3 and 5 should be directly given in the figure or the caption.
2. The meaning of the y axis and the different panels in Fig. 4 should be clarified
3. The curves in Fig. 6 should be labeled more clearly
4. Further elaboration of the deviation from the Matthiessen rule should be included.
5. More comprehensive comparison of the data with prior work on nanoporous materials should be included.
